# Telehealth Use in Geriatrics Care during the COVID-19 Pandemic—A Scoping Review and Evidence Synthesis

**DOI:** 10.3390/ijerph18041755

**Published:** 2021-02-11

**Authors:** Sathyanarayanan Doraiswamy, Anupama Jithesh, Ravinder Mamtani, Amit Abraham, Sohaila Cheema

**Affiliations:** Institute for Population Health, Weill Cornell Medicine-Qatar, P.O. Box 24144, Doha, Qatar; anj2022@qatar-med.cornell.edu (A.J.); ram2026@qatar-med.cornell.edu (R.M.); ama2006@qatar-med.cornell.edu (A.A.); soc2005@qatar-med.cornell.edu (S.C.)

**Keywords:** Coronavirus disease 19, telehealth, telemedicine, older people, gerontology, evidence mapping

## Abstract

**Introduction:** Globally, the COVID-19 pandemic has affected older people disproportionately. Prior to the pandemic, some studies reported that telehealth was an efficient and effective form of health care delivery, particularly for older people. There has been increased use of telehealth and publication of new literature on this topic during the pandemic, so we conducted a scoping review and evidence synthesis for telehealth use in geriatric care to summarize learning from these new data. **Methods:** We searched PubMed, Embase, and the World Health Organization’s COVID-19 global research database for articles published between 1 January and 20 August 2020. We included 79 articles that met our inclusion criteria. The information collected has been synthesized and presented as descriptive statistics. Strengths, weaknesses, opportunities, and threats (SWOT) have also been discussed. **Results:** The articles included in our review provide some evidence of effective provision of preventive, curative, and rehabilitative telehealth services for older people, but they highlight a greater focus on curative services and are mostly concentrated in high-income countries. We identified convenience and affordability as the strengths of telehealth use in geriatric care. Weaknesses identified include the inability of telehealth to cater to the needs of older people with specific physical and cognitive limitations. While the threats of increasing inequity and the lack of standardization in the provision of age-friendly telehealth services remain, we identified opportunities for technologic advancements driven by simplicity and user-friendliness for older people. **Conclusion:** Telehealth offers futuristic promise for the provision of essential health care services for older people worldwide. However, the extent of these services via telehealth appears to be currently limited in low and low-middle income countries. Optimizing telehealth services that can be accessed by older people requires greater government investments and active engagement by broader participation of older people, their caregivers, physicians and other health care providers, technology experts, and health managers.

## 1. Introduction

Globally, the COVID-19 pandemic has affected older people disproportionately. One estimate identifies that older people over 65 years of age have a mortality risk 100 times greater than those in younger age groups [1]. The increased vulnerability of older people to respiratory epidemics, including COVID-19, is multifactorial, driven by biological, behavioral, demographic, health care accessibility, and social determinants [2]. Among these factors, the lack of access to health care for older people is an emerging and worrying trend. Some factors contributing to this during the current pandemic are (a) an overwhelmed health sector unable to prioritize health care needs of older people, (b) the restriction of movement and lockdowns imposed as prevention measures, and the consequent challenges faced by older people trying to access health care facilities, and (c) concern and fear among older people and their caregivers of contracting COVID-19 while seeking care from health facilities. Lack of timely access to healthcare services poses an additional risk of morbidity and mortality for older people [3].

Countries and health care institutions committed to providing care to the older population have used innovative methods to ensure availability and access to health care services for older people [4], notably the use of telehealth. Prior to the pandemic, some studies reported that telehealth was an efficient and effective form of health care delivery, particularly for older people [5]. However, despite the availability of telehealth services, uptake was limited due to skepticism from both patients and the service providers. Other factors hindering the scaling up of telehealth services prior to the pandemic included concerns for patient safety, data confidentiality, absence of in-person physical examination by health providers, lack of access to technology, legal restrictions, and ambiguous health insurance coverage policies [6]. Questions were also raised about the technological literacy of older people and whether telehealth would meet the needs of older people with cognitive decline or those with auditory and/or visual impairments [7].

With the onset of the COVID-19 pandemic, a rapid scale up of telehealth services was noted globally as countries began to relax legal restrictions regarding the use of telehealth for the provision of health care services. While insurance companies in some countries had included limited telehealth services in their coverage pre-COVID-19, the scale and scope of coverage for telehealth have rapidly increased during the pandemic [8,9]. There is now a growing optimism on the global use of telehealth services in several specialties, and the field of geriatrics is no exception. Various models for the provision of telehealth services to older people have emerged during the pandemic and their strengths/opportunities highlighted [10]. However, weaknesses and threats to telehealth readiness among older people have also emerged [11]. There is a growing call to learn from the pandemic experience and contribute to the emerging knowledge on telehealth use for older people during this period. This will also improve availability, accessibility, affordability, quality, and demand generation for telehealth among older people during the ongoing pandemic and beyond.

In this article, we use the term ‘geriatrics’ to describe the whole spectrum of health care services for older people, inclusive of preventative, curative, and rehabilitative components. The term ‘telehealth’ is not currently standardized. Currently, it can include a broader perspective using such terms as e-health and digital medicine, or it can have a more specific focus such as teleneurology and telestroke. We define telehealth as the provision of health care remotely (where the patient and the health care provider do not physically meet in person). This is explained in additional detail under the sub-section ‘terminologies’ within the Methods section. Both telehealth as a specialized field in information and communication technology and geriatrics as a specialty of medicine are predominantly available in high-income countries [12,13].

It should be noted that telehealth is still in evolution, and geriatrics is a nuanced specialty given that older adults and their caregivers have unique needs [14]. The provision of telehealth services for populations who live in residential care facilities, such as nursing homes and long-term care facilities, can also be cumbersome. The provision of telehealth services in these contexts needs to be understood further. In any other population, telehealth services would largely mean service provision at the patient’s home (with the exception of specialist care from higher-level health facilities) [15].

A scoping review is an ideal tool to understand the breadth and the depth of published literature on topics that are unclear, whereas a systematic review is better to address more specific questions on well-researched topics [16]. Munn et al. (2018) provide guidance for authors on how to choose between a systematic review and a scoping review [17]. Since our intention was not to answer a single clinically relevant question but rather to identify specific concepts/characteristics relevant to telehealth use in geriatrics during the COVID-19 pandemic, a scoping review was the preferred methodology [17,18]. This approach resonates with Arksey and O’Malley’s recommendations regarding specific circumstances in which scoping reviews add value [19]. We believe that a scoping review of telehealth use in geriatrics will provide a baseline view of key concepts and characteristics in the intersection of telehealth and geriatrics, identify and analyze gaps in the knowledge base for geriatric telehealth care during COVID-19, and serve as a stimulus for future systematic reviews. We designed our scoping review and evidence synthesis to answer the research question, “What can we learn from published literature about the availability, accessibility (including demand and utilization), affordability, and quality of telehealth services in geriatric care during the COVID-19 pandemic?”

## 2. Methods

We conducted a scoping review consistent with the Joanna Briggs Institute Reviewer Manual guidance [20,21] and an evidence synthesis inspired by the Campbell Collaboration [22]. The scoping review is reported following the Preferred Reporting Items for Systematic reviews and Meta-Analyses extension for Scoping Reviews (PRISMA-ScR) [23]. The checklist is included in Appendix A. The protocol was registered in the Open Science Framework (registration DOI:10.17605/OSF.IO/26Z74) [24] on 17 September 2020.

### 2.1. Terminologies

Telehealth is an emerging field with a lack of consensus on its definition [25]. In 1997, the World Health Organization (WHO) introduced a standardized definition for telemedicine: “The delivery of health care services, where distance is a critical factor, by all health care professionals using ICT for the exchange of valid information for diagnosis, treatment and prevention of disease and injuries, research and evaluation, and for the continuing education of health care providers, all in the interests of advancing the health of individuals and their communities” [26]. ICT is defined as a “Diverse set of technological tools and resources used to transmit, store, create, share or exchange information. These technological tools and resources include computers, the Internet (websites, blogs and emails), live broadcasting technologies (radio, television and webcasting), recorded broadcasting technologies (podcasting, audio and video players and storage devices) and telephony (fixed or mobile, satellite, video-conferencing, etc.)” [27]. Telehealth includes health care services delivered by all health care professions (including education of health care professionals) in contrast to telemedicine, which is delivered by physicians only [28], although the terms are used interchangeably in the literature.

The general agreement in the field of geriatrics and gerontology is that geriatrics is a health care field dealing with the care and treatment of older persons, while gerontology has a multidisciplinary scope (involving social welfare, psychology, environment, and social systems) and deals with physical, mental, and social aspects associated with the aging process [29,30]. Since the focus of this scoping review is health care service provision for older people during the COVID-19 pandemic, we use the term ‘geriatrics’ in the manuscript.

### 2.2. Eligibility Criteria

The eligibility criteria were established a priori. We included all publications—opinions, viewpoints, original research articles, and reviews—with no geographic or language restriction.

Inclusion criteria: All publications highlighting telemedicine/telehealth use (or equivalent terms) for public health, diagnosis, treatment, prognosis determination, rehabilitation, and all other forms of health/medical care for older people (or equivalent terms) during the COVID-19 pandemic.

Exclusion criteria: Any publication not related to clinical and public health-related care of older people and/or not in the context of the COVID-19 pandemic.

### 2.3. Search Strategy

We systematically searched PubMed, Embase, and the WHO COVID-19 global research database for articles published between 1 January and 20 August 2020 with no geographic and language limitations, using a combination of keywords as well as a controlled vocabulary. The start date of the search strategy (1 January 2020) corresponded to the earliest date when literature in the COVID-19 context was likely to have been published (the first cluster of cases were reported in Wuhan, China, 31 December 2019) [31]. 20 August 2020 was the last day we conducted the search in all three databases.

Good practice in conducting a scoping review involves searching at least two online databases [20,21]. We included a third database to improve the comprehensiveness of our search. Notably, Pubmed and Embase are the preferred databases for searching primary studies as part of scoping and systematic reviews [32]. Pubmed, as a proxy to MEDLINE, provides extensive coverage to the point where some authors have suggested that doing a Pubmed search alone may be sufficient for reviews [33]. Embase serves as a good complement to Pubmed and is known to produce unique references along with coverage of second-tier European and Asian journals [32]. With the inclusion of several databases in one online database, the WHO’s COVID-19 global research database increases the prospects of newly published articles that may not have yet been indexed in Pubmed and Embase [34].

We built the search strategy step-by-step according to the JBI manual for scoping review [20,21]. Firstly, a limited search was conducted on Pubmed and Embase for relevant articles. The initial search was followed by an analysis of the text words contained in the title and abstract of retrieved articles and of the index terms used to describe the articles. This helped us develop the three concepts for our search strategy: (a) COVID-19, (b) telehealth/telemedicine, and (c) old age. These concepts and the final choice of our databases were discussed and agreed upon in consultation with a senior librarian. Thereafter, an initial set of key terms was developed for our search strategy through a systematic brainstorming process involving all authors, each of whom has significant experience publishing scoping and systematic reviews together. On the advice of the senior librarian, we further compared our key terms with the terms included in the search strategy compilations done by the Medical Library Association [35]. The final search strategy was validated by the librarian and the senior authors. We then carried out a second search using all identified keywords and index terms in the three databases. It is to note that Pubmed (MeSH) [36] and Embase (EMTREE) [37] use specific systems of classification of key words. Our search included words borrowed from the respective systems of the classification used by these databases. However, the WHO global COVID-19 database permits the use of keywords only, and our search strategy for this database was drawn from our search strategy in Pubmed and Embase. Thirdly, the reference list of articles in the included full-texts was searched for additional articles. The detailed search strategy for each of the databases is available in Appendix A.

### 2.4. Selection and Extraction

The final search strategy was deployed jointly by S.D. and A.A. in the three databases. The title and abstract of all identified articles were imported into Rayyan [38], the online systematic review software. S.D. and A.A. removed the duplicates together following consultation. S.D. screened the title and abstracts of the identified articles. A.A. checked the excluded articles and confirmed the correct application of exclusion criteria. A.A. checked an additional 10% of the included articles to confirm the application of the screening criteria. S.D., A.A., and A.J. discussed the progress together and had shared access to the database of included and excluded articles in Rayyan. S.D. mobilized all full texts, and A.J. complemented the work by mobilizing missing full texts through inter-library loan. After mobilizing all full texts, S.D. transferred all articles to Endnote X9 [39]. S.D., A.A., and S.C. met and revalidated the inclusion and exclusion criteria that had been established a priori by all authors. S.D. screened full text of all articles based on the set eligibility criteria. A.J. checked all excluded articles and discrepancies were resolved through discussion. S.D. and A.A. developed a standardized data extraction Microsoft Excel template to tabulate the following information from the articles: author, title, uniform resource locator (URL) link, publication type, country of the first author, country of focus, name of the journal, terminologies used, definitions, purpose, health discipline, specific theme/topic, opportunities, challenges, lessons learned, and recommendations. S.D. extracted the data, and A.J. independently extracted data from 10% of the included studies to check and ensure correctness and completeness of the data extracted by S.D. Since no discrepancies were noted, and the need to involve a third reviewer did not arise, it was agreed between team members for A.J. to check the correctness and completeness only of the data charted by S.D. A few minor discrepancies were observed that were resolved through discussion between S.D. and A.J. All team members were kept informed during regular update meetings, and no further discrepancy resolution was required.

### 2.5. Data Analysis and Synthesis

Collected data was primarily synthesized and presented as descriptive statistics (frequencies and proportions). We used Microsoft Excel to tabulate and analyze the data. The analyzed data is presented as tables in the Results section. In addition, we also provide a visual representation of the domains for which varying evidence is available for telehealth use in geriatrics during the COVID-19 pandemic. This is a useful adjunct to a scoping review to best demonstrate the breadth and depth of evidence available for the use of telehealth in geriatrics during the pandemic [40].

We compiled the key opportunities, challenges, lessons learned, and recommendations from the included articles and qualitatively summarized them in tables. We used the Strengths (S), Weaknesses (W), Opportunities (O), and Threats (T) framework, also called the SWOT framework, to better contextualize the use of telehealth in geriatrics during the pandemic. Where the authors of the articles presented their findings/views from the perspective of older people/caregivers and physicians/other health care providers, we categorized them under strengths and weaknesses of telehealth in geriatrics. Articles that discussed technology and the health system/governance perspectives were categorized under opportunities and threats. We specifically looked for factors that improved/inhibited availability, accessibility (including demand and utilization), affordability, and quality of telehealth services provided in geriatric care. The SWOT framework was chosen as it is useful not only to group facilitatory factors and barriers in a systematic manner but also provides a good template to direct recommendations to specific stakeholders [41].

## 3. Results

### 3.1. Selection of Articles

After duplicate removal, our search strategy identified 548 articles relevant to our review. Of these, 446 articles were excluded at the stage of title and abstract screening and another 30 articles during full-text screening. Among the 30 articles excluded after full-text screening, twelve articles dealt with issues such as electronic health records, artificial intelligence, and computational analysis, which are not traditionally considered to be part of telehealth; twelve articles used information technology to conduct research; two articles discussed past pandemics; two were studies carried out before the onset of the COVID-19 pandemic; one article was a simulation study to assess adverse drug events; one article was a study protocol. Our supplementary search of the reference lists from the included articles yielded an additional seven articles relevant to our review. A total of 79 articles were finally included in our scoping review. The PRISMA flowchart representing the article selection process is included in Figure 1.

### 3.2. Characteristics of the Included Articles

The 79 articles included in our review were published in 44 journals. This journal that published the highest number of articles was the *Journal of American Geriatric Association* (9/79; 11.4%), followed by the *Journal of the American Directors Association* (7/79; 8.9%), the *American Journal of Geriatric Psychiatry* (6/79; 7.6%), the *Journal of Gerontological Social Work* (4/79; 5.1%), and the *Journal of Medical Internet Research* and its sister journals (4/79; 5.1%). For 78/79 (98.7%) publications, we could identify the publication category in the journals. Among these publications, 15/78 (19.2%) were categorized as empirical research, and 5/78 (6.4%) were reviews. The other articles (58/78; 74.3%) belonged to various categories, including letters to the editor (20/78; 25.6%); COVID-19-related special articles (12/78; 15.4%); editorials (6/78; 7.7%), commentaries, opinions, perspectives, viewpoints, etc. Except for two articles in Spanish, all other articles (77/79; 97.5%) were in English.

The articles included in our review discussed telehealth for older people in the context of 20 countries. Seventeen of these 20 countries belonged in the World Bank high-income country category, two in the upper-middle-income category, and one in the lower-middle-income category. Grouping the countries according to the World Health Organization (WHO) regions, 18/20 (90%) countries were found to be divided among Europe (9/20; 45%), Western Pacific (5/20; 25%), and the Americas (4/20; 20%). There was only one country each (1/20; 5.0%) from the Eastern Mediterranean and South East Asia regions.

The majority of the articles focused on the United States of America (34/79; 43%) followed by Canada (7/79; 8.9%), Italy (5/79; 6.3%), Spain (5/79; 6.3%), and the United Kingdom (5/79; 6.3%). The geographic focus of the articles in our review and the country groupings according to the World Bank income categories [42] and WHO regions [43] are included in Appendix A.

Thirty-three articles indicated the setting in which telehealth was applied: 16/33 focused on the provision of telehealth in private homes (48.5%), 14/33 in assisted homes (42.4%), and 3/33 (9.1%) in hospital settings. The majority of the articles (52/79; 65.8%) covered specific aspects (preventative, curative, or rehabilitative) of healthcare, while the remainder discussed the general application of telehealth for older people. Of the articles examining telehealth for specific aspects of healthcare: 32/52 (61.5%) discussed curative services for older people; 15 articles (28.8%) examined health promotion (maintaining a positive lifestyle, including virtual social connectedness); three articles (3/52; 5.8%) focused on COVID-19 related services; and one article (1/52; 1.9%) discussed rehabilitative services for older people. Only one article (1/52; 1.9%) discussed telehealth use in a purely (primary) preventative context.

In our review, 44/79 (55.7%) articles focused on eight medical specialties, one on dentistry (1/79; 1.3%), and 35/79 (44.3%) on general geriatric health care services for older people. The top three specialties discussed in the articles were Psychiatry (14/44; 31.8%), Neurology (10/44; 22.7%), and Preventive Medicine (6/44; 13.6%). Discussion on sub-specialties and/or specific disease conditions were found in 24/79 (30.4%) articles. Dementia (9/24; 37.5%) and frailty (5/24; 20.8%) were the most common conditions for which telehealth services were used. The medical specialties/subspecialties and specific disease conditions (*in italics*) discussed in the articles are depicted in Figure 2.

In the 79 articles, 49 terminologies were used to refer to telehealth-related services. The only terminology specific to geriatrics we identified was ‘gerontechnology’. Although no clear definition for this term was provided in the article [44], the international gerontological society defines the term gerontechnology as “designing technology and environment for independent living and social participation of older persons in good health, comfort, and safety” [45]. The list of other terminologies and definitions are included in Appendix A.

We mapped the evidence for telehealth use in older people using the Strengths, Weaknesses, Opportunities, and Threats (SWOT) framework (Table 1). Strengths and opportunities were extracted from 28 articles and weaknesses and threats from 33 articles. Among the articles that discussed strengths and opportunities related to telehealth use, eight discussed the viewpoints of physicians and other health care providers, and six each discussed technological perspectives, health system/governance perspectives, and the perspectives of older people and their caregivers. In summary, the ease of seeking holistic health care and risk avoidance while attending health facilities are strengths from the viewpoint of older people. From a physician and health care provider perspective, telehealth strengths were reported as facilitation of the provision of holistic and personalized health care (for conditions such as dementia) to older people in remote locations and others within their environment of comfort (home). This is deemed to lead to minimizing infection risk from health facilities. In nursing homes, telehealth was found to promote connectedness between families and older people, and a reduction in the level of caregiver involvement generally required when accompanying older people to health facilities. The opportunities identified include making the technology element more user friendly for both older people and health care providers.

While discussing weaknesses and threats, most articles (13/33; 39.4%) focused on the perspective of health system/governance, followed by physicians and health care providers (9/33; 27.3%), technology, 8/33 articles (24.2%) highlighted the challenges faced by older people in using telehealth services and technology (6/33; 18.2%). Weaknesses reported include the limitation of telehealth in providing health care services to older people with cognitive challenges, health care providers having limited skills in using telehealth, and situations where access to technology is limited and/or where maintenance of technological equipment may be daunting for older people. Lack of privacy in some home settings and the inability to perform a physical examination by health care providers were also cited as compromising the quality of care for patients. The threats to the progress of telehealth use in geriatrics include lack of standardization; human and technological errors; a lack of government policies to support older people, especially those who are poor, thus widening inequality; and prevailing ageism that may lead to non-prioritization of older people in the scale up of telehealth services.

For evidence synthesis, we grouped the articles according to the publication type (as a proxy to the quality of evidence) against the SWOT analysis for telehealth use in geriatrics (Figure 3). Given the scoping nature of our review, a visual map has been prepared to (a) demonstrate a quick overview of the three categories of evidence—opinions (including opinions/reports/editorials/recommendations/perspectives as published in the journals), primary studies (single studies and case studies), and reviews (literature and scoping reviews only, as there were no systematic reviews), (b) provide a visual summary of the four common scopes of service—namely (i) preventative (including screening, assessments, triaging, promoting a healthy lifestyle and enhancing social connectedness) (ii) curative (including treatment and follow-up to treatment services) (iii) rehabilitative (including the provision of physiotherapy, prosthetic management, and bereavement services) and (iv) general healthcare services with no specific focus and (c) convergence of the categories of evidence and the scope of the service. The number of circles in the visual map represent the number of articles of that type.

The visual map demonstrates the general preponderance of opinions across all scopes of service (preventative, curative, rehabilitative, general/non-specific), with the predominant focus of telehealth use being on curative and a very minimal focus on rehabilitative services. Articles reporting strengths and weaknesses (from the point of view of older people and physicians) predominate over articles reporting opportunities and threats (from the point of view of technology and health system/governance). The list of articles grouped according to the various categories has been presented in Appendix A.

Regarding recommendations for improving telehealth services for older people, 34/79 (43.0%) of the articles provided these. These recommendations are presented in Table 2. We observed some similarities in the recommendations provided by the articles included in our review: five articles (5/34; 14.7%) called for greater participation of older people in ICT training; four articles (4/34; 11.8%) advocated for older people to make informed decisions based on understanding the advantages and disadvantages of telehealth taking into consideration their specific context/situation; and five articles (5/34; 14.7%) demanded a proactive role for older people in the telehealth standardization process, inclusive of open discussions with their service providers. Recommendations for health care professionals were found in five articles (5/34; 14.7%) supporting personal skills improvement, empowering their patients, and adapting existing clinical tools for digital use. Four articles (4/34; 11.8%) discussed the need for technology to meet the needs of older people with cognitive impairment, while others (8/34; 23.6%) called for simplification and greater security assurance of technology. Thirteen articles (13/34; 38.2%) proposed older people-friendly policies and systems-strengthening approaches that would lead to improvements in the availability, accessibility, affordability, and quality of telehealth services.

## 4. Discussion

Our scoping review and evidence synthesis summarizes data and information from 79 articles pertaining to telehealth use for the provision of geriatric care. During the COVID-19 pandemic, physical distancing for older people has been recognized as a mechanism to mitigate spread in this high-risk category and thus been implemented in several countries worldwide. This has also allowed more flexibility for telehealth use in geriatric care.

### 4.1. Availability, Accessibility, Affordability, and Quality of Telehealth in Geriatric Care during the COVID-19 Pandemic

COVID-19 has demonstrated the value of telehealth in providing geriatric care during the pandemic. Relying on telehealth to provide continuity of geriatric care services and avoid the risk of contagion by reducing the need for visiting health care facilities has been shown to be feasible during the pandemic [104]. It is noteworthy that a majority of the articles in this review were written in the context of high-income countries, notably the United States of America (USA), and were published predominantly in geriatrics-related journals published in those countries. This is similar to other studies, which have found that the vast majority of published articles on telehealth were from studies carried out in high-income countries, and a vast majority of them were from the USA [9]. In high-income countries, access to and use of telehealth for older people has improved during the pandemic. This occurred due to the relaxation of legal restrictions for providing health care and the inclusion of telehealth as a reimbursable service by insurance companies in countries such as Australia [65]. The same cannot be said about low- and low-middle-income countries from which the telehealth literature is sparse.

A wide application of telehealth services has been seen during the pandemic in both home and long-term care settings in which older people live. In long-term care settings, telehealth has been used as an adjunct to provide collaborative support for getting advice from multiple specialists concurrently. The articles included in our review provide some evidence of the application of telehealth in providing a whole spectrum of health care services, including preventative, curative, and rehabilitative services but with a greater focus on curative services.

Among the curative services provided to older people during the pandemic, our review finds that there is more literature on telehealth application for neuropsychiatry services than other specialty services. This finding reflects the fact that frailty and dementia are the more common conditions managed by telehealth services. There seems to be a limited number of other medical specialties using telehealth to provide services to older people. Worldwide though, the use of telehealth in other specialties such as dermatology, pathology, and radiology has been well documented as part of non-geriatric health care [28]. There is much potential to expand various telehealth services to older people in the future.

Only a small proportion of the articles identified in our review were based on empirical research, so there is clearly a need for additional research to generate good quality evidence on telehealth use by older people. Nevertheless, the available evidence provides invaluable information. Prior to COVID-19, telehealth was not widely available to older people, citing a lack of capacity on their part to navigate the technology needed for its use [105]. However, a majority of the articles in our review report an increasing interest and uptake of telehealth by older people since the onset of the COVID-19 pandemic. When telehealth is provided in an age-friendly manner with active collaboration between older persons and their health care providers, we are likely to see an increased service uptake [105]. This is a significant finding.

### 4.2. Strengths, Weakness, Opportunities, and Threats for Telehealth Use in Geriatric Care during the COVID-19 Pandemic

In our SWOT analysis, strengths identified for telehealth use in geriatric care are the convenience and affordability for older people. The weaknesses identified showcase that telehealth as a field must evolve and adapt to meet the needs of older people, specifically those with physical and cognitive limitations. Also, the gap in telehealth knowledge and capacity for use by health care providers must be addressed. The threats focus on inequity and the lack of standardization in the provision of age-friendly telehealth services. The articles included in our review identify opportunities primarily in the technological advancements driven by simplicity and user friendliness. All these identified areas offer broad scope for future exploration.

### 4.3. Implications for Practice and Research Gaps

Given global concerns on the quality of publications that emerged since the onset of the pandemic [106,107], our decision to exclude gray literature and restrict our search to peer-reviewed articles only can be seen as the most appropriate approach to synthesize available evidence. Our scoping review found that optimizing telehealth services for older people requires broader engagement with broader participation by older people, their caregivers, physicians, and other health care providers, as well as technology experts and health managers. Our scoping review found a limited number of articles that can answer questions on the safety, utility, scalability, cost effectiveness, and demand for telehealth use in geriatrics. Therefore, there is a clear need for additional research in these areas during the pandemic and beyond. There is also a need to address the lack of standardization of telehealth terminologies used in the geriatric context [25].

## 5. Limitations

Though our scoping review is comprehensive in exploring the subject during the pandemic, it has limitations. By virtue of being a scoping review, we have not done a quality assessment of the articles included in our review. That being said, since all articles are from peer-reviewed journals, the quality can be assumed to be reasonably assured. This is also our reasoning for excluding gray literature (inclusive of preprints), which may be identified as a limitation of our review. Ideally, screening and data extraction should be done in duplicate. But owing to the logistic constraints imposed by the pandemic with new working arrangements, quarantine, and isolation measures in country, the volume of publications that were being generated on the topic, and the need to generate useful data to guide quality research, we opted for the second-best option of one reviewer screening and extracting the data and the other checking to address any discrepancies. This helped us to ensure a wider coverage of peer-reviewed articles, including letters to the editor, opinions, commentaries, and editorials. Their inclusion in scoping reviews is recommended to get a better perspective of the breadth of publications on the topic [17]. We duly followed all the mandatory steps recommended by Arksey and O’Malley in their methodological framework for scoping reviews. However, we did not pursue the optional step of consultation, given the pandemic-related limitations indicated above [19].

## 6. Conclusions

The COVID-19 pandemic has posed a major challenge to humanity. However, this crisis has allowed us to explore and better understand the use of telehealth for older populations. Telehealth offers futuristic promise for the provision of essential health care services to older people. Currently, the extent of these services via telehealth appears to be limited in low and low-middle income countries. A greater commitment to and resource allocation for telehealth services are needed in these countries to allow older people to avail of and benefit from these services. Social responsibility also rests on middle- and high-income countries with the available technologies for the provision and sharing of telehealth services with socio-economically disadvantaged communities. Countries already advanced in geriatrics and telehealth services should continue to invest in innovation and robust research to ensure the adoption of age-friendly telehealth services that not only meet the care needs of older people and their caregivers but also allay any concerns that may exist.

## Figures and Tables

**Figure 1 ijerph-18-01755-f001:**
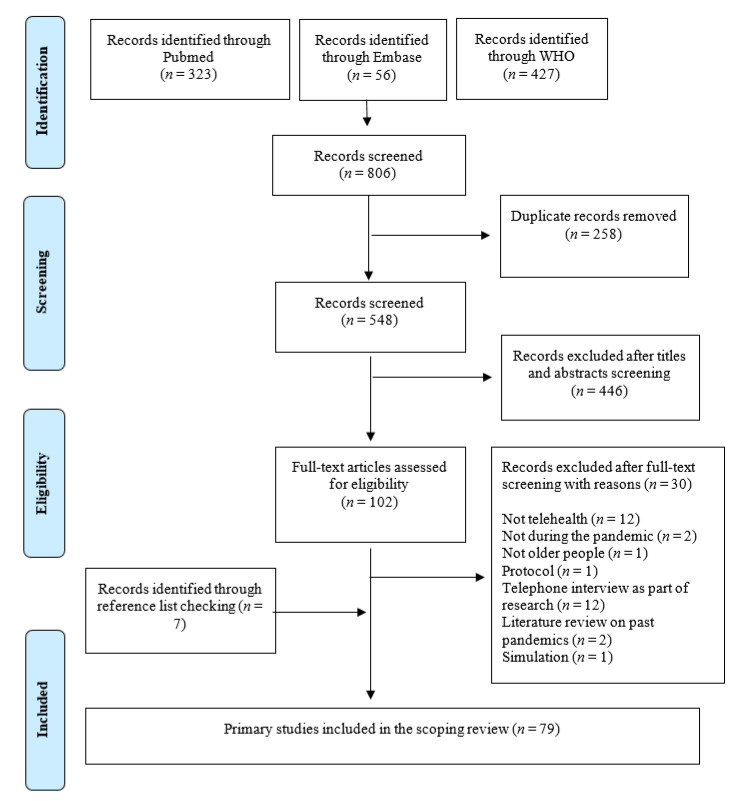
PRISMA 2009 flowchart of the scoping review’s inclusion.

**Figure 2 ijerph-18-01755-f002:**
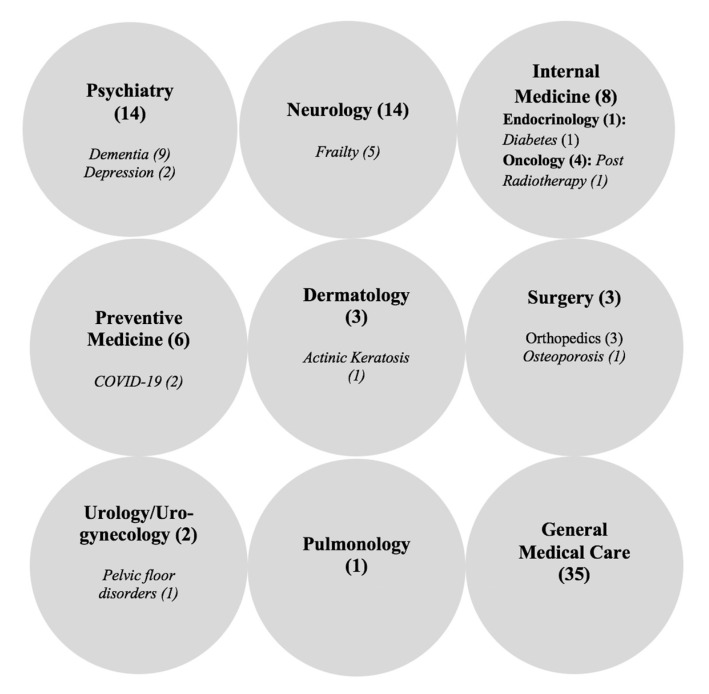
Medical specialty and sub-specialty/specific disease conditions.

**Figure 3 ijerph-18-01755-f003:**
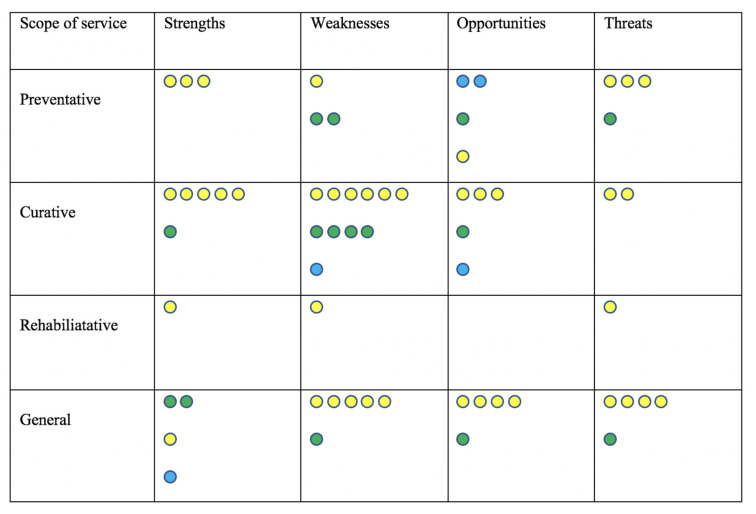
Visual map of type of articles/evidence. **Preventative**—Screening/Assessment/Triage/Lifestyle/Social connectedness; **Curative**—Treatment/Follow-up **Rehabilitative**—Physiotherapy; prosthetics management and bereavement support; **General**—No definitive focus; **Green circles**—Single studies/Case study; **Blue circles**—Literature and Scoping reviews; **Yellow circles**—Opinions/Reports/Editorials/Recommendations/Perspectives—Number of small circles represent the number of articles of specific type/category.

**Table 1 ijerph-18-01755-t001:** The use of telehealth for older people—SWOT analysis.

**S—Strengths** Avoid over-crowding in health facilities [46] Reduction in long distance travel [46] Minimize risk of serious events such as falls [46] Improve resilience and well-being [47,48] Able to perform an assessment in the living (ecological) environment [46,49,50,51] Service patients from rural communities [52,53] Reduce secondary and tertiary infections [54] Reduce loss to follow up [55] Holistically manage dementia [52,56] Personalized management including reminders [53,57] Reduce caregiver involvement, including nursing home staff [52,58] Promote social connectedness among nursing home residents in particular [44,59]	**W—Weaknesses** Difficult to treat patients with cognitive impairment, visual acuity issues, and hearing problems [56,60,61,62,63,64] Gauging patient comprehension by providers [65,66] Limitation in physical examination including gait assessment [66,67,68] Lack of older people’s inclusion in the design and user-testing of telehealth interventions [56,61] Greater risk of treatment withdrawal and increased risk of hospitalization [69] Management of video connectivity problems telehealth platforms [70] Maintenance of equipment, including its sanitization and the associated extra burden [55] Lack of privacy for the older person [71] Risk of missing out on clues of elder abuse due to the lack of privacy [70] Lack of familiarity of health care professionals with telehealth platforms [65,72]
**O—Opportunities** Free communication platforms [44,59,73,74,75,76] Readily available web-based training [77,78,79] Standardized documentation and real time reporting to improve quality of care [80] Digital photographs and asynchronous sharing to circumvent connectivity issues [77] Wearable devices, remote monitoring sensors and other technologies as early warning tools [59,81] Automatic speech analysis for diagnosis and monitoring of dementia [82] Enhanced integration of specialty expertise care of nursing home residents [70,83] Possibility of daily community collaborative rounds involving multiple services providers in nursing homes [84] Technology that is easy to understand pertaining to different interfaces, passwords, and maintenance [56,70]	**T—Threats** Ambiguous/technical jargon for descriptive terms [85] Large variability of available telehealth platforms [70] Lack of sustained insurance reimbursement [11] Digital divide for some due to lack of equipment, limited literacy, and lack of assistance [56,63,72,86,87,88,89,90,91,92,93] Ageism/stigma leading to de-prioritization of older people [94,95] Failure to include older people in standardizing telehealth [56] Lack of tested clinical tools for telehealth use [65,96] Technical failures and patient dropout [89,97] Inaccuracy in telephone consultation [98] Difficulty for regulators in monitoring and ensuring equitable quality of care [52,65,84]

**Table 2 ijerph-18-01755-t002:** Recommendations for improving telehealth services for older people.

**Older People and Caregivers** Participate in communication technology training [61,73,85,91,94] Make an informed decision to use telehealth for health care services assessing the pros and cons [44,52,68,70] Discuss concerns with health care providers and explore solutions to mitigate concerns [70] Be part of blended communities that bring together online and in-person activities [91,99] Be involved in the process of standardization of telehealth use for geriatric care [70,85]	**Physicians and Other Health Care Providers** Get trained in assessing patients using video conference [52] Convert current in-person screening and diagnostic tools into digital versions [52] Dedicate time in asking patients questions about concerns and barriers to accessing technology [93] Redirect patients to educational community resources for telehealth use when necessary [95]
**Technology** Develop automatic speech analysis to diagnose and monitor dementia [82] Improve technology to accommodate age-related sensory and cognitive impairment [46,63,64,94] Replace ambiguous/technical jargon with easily understandable terms [64,100] Develop larger touchscreen tablets to make visual acuity less of a barrier [64,94] Use a simple and timely back-up process in the event of equipment or connectivity failure [70,97] Address hacking risk by ensuring the use of secure software [70]	**Governance/Health Systems** Provide educational outreach to support older adults to use digital devices [87,91,94] Ensure sanitization of telemedicine equipment [101] Sustain reimbursement for telehealth services beyond the COVID-19 pandemic [102,103] Integrate telehealth within the training curricula for both health and social care professionals and practitioners [65,70] Provide equipment such as tablets, laptops, or devices that can connect to the TV for older people and caregivers [97] Ensure free internet service to all, including older people [91,94]

## Data Availability

The datasets generated during and/or analyzed during the current study are available from the corresponding author on reasonable request.

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
