# Peer review of "Telehealth Use in Geriatrics Care during the COVID-19 Pandemic—A Scoping Review and Evidence Synthesis"

_ijerph, 2021, doi:10.3390/ijerph18041755_

Round 1

Reviewer 1 Report

Thank you for taking on the initiative of reviewing telehealth implementation during the COVID-19 pandemic as applied to geriatric care.  I enjoyed the premise of the paper and have the following suggestions to ensure it can be the best manuscript possible:

1.  terminology.  Please further define (with citations) the key variables used in the study.  Specifically:  telehealth vs. telemedicine, gerontology/geriatric care.  Terms are used or discussed in varying ways throughout the manuscript and this should be more streamlined.

2.  please further describe/define the purpose of a scoping review, as compared to a systematic review (and/or follow-on research to be conducted upon completion of the scoping review).  In other words, please further discuss the reasoning for a scoping review, versus other research methods available.

3.   the eligibility criteria section further highlights the issue of telehealth vs. telemedicine terminology.  Please discuss how/why both of these terms were used as concepts in the study, versus one or the other.  In reading the manuscript, it seems the primary focus is telemedicine - but that's simply my assumption at this point.

4.  database selection - please further discuss how/why each research database was selected and used in the study.  Next - a discussion is warranted to provide the reader information as to why each database was queried using varying search criteria (specifically terminology as noted in an appendix provided).

5.  please qualify the reasoning for the data range search criteria.

6.  the search strategy (validation by peers with experience...) as well as the three search concepts is very weak.  This section requires significant explanation and further details to solidify the reader's confidence in the search parameters and qualifying manuscripts in the sample.  Please discuss how these exact three search concepts were decided upon and justify/support the search criteria used (especially if different criteria 'words' between research databases).

7.  the selection criteria relies too much on any single research team member.  This introduces significant bias into the manuscript - even if checked afterward by another team member.  This lack of individual review at the team member level and then collaboration afterward to address potential selection bias discredits a significant amount of the study.  If there were more teamwork and collaboration in the search criteria development, selection, etc - this would be a much different (overall) review outcome.

8.  Figure 1 is incomplete/parts running-off the page/figure that are not readable.  Reasons for excluding after full-text reading needs to be further addressed in the manuscript and not just noted on the figure.

9.  Table 1 could possibly be interpreted more as a figure (visual, Venn diagram, etc) versus listing the data in table format.

10.  Table 2:  suggestions include - a) including the frequency of citations/mention for each line-item to support the reader's understanding of frequency of mention in the literature; b) include citations to support the findings in this table (versus listing in an appendix and not tied-into the table information); c) a summary of each variable (SWOT) as applied to the study's research topic is necessary to assist in interpretation.

11.  Figure 2 requires a detailed legend to assist in interpretation.  Additionally, a significant of explanation and discussion on the 'synthesis' section accompanying this figure is lacking.

12.  Table 3 requires direct citations (similar comment to Table 2 above) and further discussion as to the identified concepts.  If individually listed from individual citations, broader concepts should be identified by the researchers and then supported by the literature from Table 2.

13.  Please greatly expand the discussion section, to include sub-headings that map to the overall concepts drawn from the scoping review and related synthesis mentioned.

14.  the limitations section further supports comments above regarding the review's methodology.  Why not address selection bias (limited reviewers) before completing the task?  Why not address the quality of articles on the front-end, versus including letters to the editor/etc and then listing as a study weakness?  Please further address these method shortcomings as related to 'logistic challenges' mentioned here.  While it discourages acceptance of the paper, it may be further explained to address pandemic situations/limitations and still support contributions to the field if interpreted as such.

I enjoyed the paper.  I do believe significant efforts in a revision/resubmission will result in a much stronger paper that contributes to the field.  Thank you for the opportunity to review.

Author Response

Response to Reviewer:

Reviewer 1:

Thank you for taking on the initiative of reviewing telehealth implementation during the COVID-19 pandemic as applied to geriatric care.  I enjoyed the premise of the paper and have the following suggestions to ensure it can be the best manuscript possible:

Response: All authors would like to thank the reviewer for the kind words about the premise of our paper. We would also like to thank the reviewer for the time and effort spend in providing us these valuable comments, observations, and suggestions to strengthen our paper. We have addressed these in the revised version.

  1. Please further define (with citations) the key variables used in the study.  Specifically:  telehealth vs. telemedicine, gerontology/geriatric care.  Terms are used or discussed in varying ways throughout the manuscript and this should be more streamlined

Response: We have now included a section ‘terminologies’ under Methods. We have clarified and added citations to the terminologies used and have streamlined the use of these terminologies throughout the paper.

  1. please further describe/define the purpose of a scoping review, as compared to a systematic review (and/or follow-on research to be conducted upon completion of the scoping review).  In other words, please further discuss the reasoning for a scoping review, versus other research methods available.

Response: We have now included text in the introduction section providing reasoning for our preference to use the methodology of a scoping review as compared to a systematic review. We have used the below citation as the basis for our choice.

Munn Z, Peters MDJ, Stern C, Tufanaru C, McArthur A, Aromataris E. Systematic review or scoping review? Guidance for authors when choosing between a systematic or scoping review approach. BMC Medical Research Methodology. 2018;18(1):143.

  1. the eligibility criteria section further highlights the issue of telehealth vs. telemedicine terminology.  Please discuss how/why both of these terms were used as concepts in the study, versus one or the other.  In reading the manuscript, it seems the primary focus is telemedicine - but that's simply my assumption at this point.

Response: In addition to the section on terminologies now introduced based on comment 1, we have also clarified the lack of standardized use of telehealth and telemedicine terminologies. We have now included references to substantiate the point that telemedicine is a focused sub-field within the larger field of telehealth. The focus of our review was telehealth to cover the entire spectrum of technology-enabled services. However, our review and evidence mapping does find that a majority of the work is in the sub-field of telemedicine with a predominant focus on curative services. See below the key reference to substantiate this point:

Fatehi F, Wootton R. Telemedicine, telehealth or e-health? A bibliometric analysis of the trends in the use of these terms. Journal of telemedicine and telecare. 2012;18(8):460-4.

  1. database selection - please further discuss how/why each research database was selected and used in the study.  Next - a discussion is warranted to provide the reader information as to why each database was queried using varying search criteria (specifically terminology as noted in an appendix provided).

Response: We have now included a full paragraph in the Methods section explaining the build-up of the search strategy including the reasoning for the choice of the three databases used in the study.  We also further elaborate on the key words systems of classification in Pubmed and EMBASE – MeSH and Emtree respectively.

  1. please qualify the reasoning for the data range search criteria.

Response: We have now included the justification for the date range in the search criteria within the Methods section. The start date (Jan 1, 2020) corresponds to the earliest date when publications on the topic in the context of the COVID-19 pandemic were likely to have arisen (the first cluster of cases reported on 31 December 2019) and the end date (20 August) was the date on which the search was concluded in all three databases.

  1. the search strategy (validation by peers with experience...), as well as the three search concepts, is very weak.  This section requires significant explanation and further details to solidify the reader's confidence in the search parameters and qualifying manuscripts in the sample.  Please discuss how these exact three search concepts were decided upon and justify/support the search criteria used (especially if different criteria 'words' between research databases).

Response: We would like to thank the reviewer again for suggesting this important value addition to our paper. As indicated in our response to comments 4 and 5, we have significantly expanded and elaborated the systematic steps we took in building our search strategy.

  1. the selection criteria relies too much on any single research team member.  This introduces significant bias into the manuscript - even if checked afterward by another team member.  This lack of individual review at the team member level and then collaboration afterward to address potential selection bias discredits a significant amount of the study.  If there were more teamwork and collaboration in the search criteria development, selection, etc - this would be a much different (overall) review outcome.

Response: We would like to assure the reviewer of the highly collaborative process undertaken in conducting the scoping review within the logistic constraints imposed by the pandemic, limiting the ability to physically collaborate at all stages. We have now greatly elaborated on how team members engaged with each other at each stage of the review process and maintained a high level of collaboration to minimize bias.

  1. Figure 1 is incomplete/parts running-off the page/figure that are not readable.  Reasons for excluding after full-text reading needs to be further addressed in the manuscript and not just noted on the figure.

Response: We would like to thank the reviewer for this observation and suggestion. The running-off of the figure occurred due to the journal (auto) formatting our submitted manuscript. We have now corrected it in the updated version. We have also now included the reasons in the text for the exclusion of articles following full-text screening.

  1. Table 1 could possibly be interpreted more as a figure (visual, Venn diagram, etc) versus listing the data in table format.

Response: We have now converted the table into a figure to ensure ease of readability.

  1. Table 2:  suggestions include - a) including the frequency of citations/mention for each line-item to support the reader's understanding of the frequency of mention in the literature; b) include citations to support the findings in this table (versus listing in an appendix and not tied-into the table information); c) a summary of each variable (SWOT) as applied to the study's research topic is necessary to assist in interpretation.

Response: We have now included the citations into the table directly – thus providing the reader an understanding of the frequency of citations including the actual citations themselves. We have also included a detailed summary of the variables in the Results section.

  1. Figure 2 requires a detailed legend to assist in interpretation.  Additionally, a significant explanation and discussion on the 'synthesis' section accompanying this figure is lacking.

Response: We have now included a significant explanation of the figure in the section accompanying it.

  1. Table 3 requires direct citations (similar comment to Table 2 above) and further discussion as to the identified concepts.  If individually listed from individual citations, broader concepts should be identified by the researchers and then supported by the literature from Table 2.

Response: We have now included the citations into the table directly. We have identified broader concepts to group the recommendations and have included them in the text within the Results section. We have further amplified this in the Discussion section in line with the response to comment 13 below.

  1. Please greatly expand the discussion section, to include sub-headings that map to the overall concepts drawn from the scoping review and related synthesis mentioned.

Response: We have now included sub-headings in our discussion section and elaborated relevant sections to signify the overall concepts drawn from the scoping review and evidence synthesis.

  1. the limitations section further supports comments above regarding the review's methodology.  Why not address selection bias (limited reviewers) before completing the task?  Why not address the quality of articles on the front-end, versus including letters to the editor/etc and then listing as a study weakness?  Please further address these method shortcomings as related to 'logistic challenges' mentioned here.  While it discourages acceptance of the paper, it may be further explained to address pandemic situations/limitations and still support contributions to the field if interpreted as such.

Response: We have now clarified the limitation section further to include the time and logistical constraints imposed by the pandemic.

I enjoyed the paper.  I do believe significant efforts in a revision/resubmission will result in a much stronger paper that contributes to the field.  Thank you for the opportunity to review.

Response: All authors would like to thank the reviewer again for the time and for the constructive feedback provided to strengthen the manuscript. We believe that the manuscript has been significantly strengthened with the comments and in our efforts to resolve them.

Reviewer 2 Report

Thank you for the opportunity to review your work. Your study addresses an important (but implicit) research question and has high value. Please see the following  recommendations:

Introduction:

  1. Highlight what is unknown about telehealth in geriatrics during the pandemic and why the review questions/objectives lend themselves to a scoping review approach.
  2. Provide an explicit statement of the research questions and objectives being addressed with reference to their key elements (e.g., population or participants, concepts, and context).

Results

  1. Tables: Please, align left the content data for easier reading.
  2. Figure 2: In its description, the authors need to include a description of the colored circles.

Author Response

Response to Reviewer:

Reviewer 2:

Thank you for the opportunity to review your work. Your study addresses an important (but implicit) research question and has high value.

Response: We would like to thank the reviewer for the kind words about the premise of our paper. We would also like to thank the reviewer for the invaluable comments and suggestions to strengthen the paper. We have given serious consideration to all the comments and have addressed them in our revised version.

Please see the following  recommendations:

Introduction:

  1. Highlight what is unknown about telehealth in geriatrics during the pandemic and why the review questions/objectives lend themselves to a scoping review approach.

Response: We have now highlighted this in the Introduction section.

  1. Provide an explicit statement of the research questions and objectives being addressed with reference to their key elements (e.g., population or participants, concepts, and context).

Response: We have now made this explicit in the Introduction section.

Results

  1. Tables: Please, align left the content data for easier reading.
  2. Figure 2: In its description, the authors need to include a description of the colored circles.

Response: We have now organized the tables and figure 2 (now figure 3 in the updated manuscript) accordingly.

Reviewer 3 Report

The manuscript is nicely written and highly relevant for the research community in telehealth.

I just have some suggestions to enhance the transparency of the result in the attached document.

Furthermore, in the discussion section, you make clear that gray literature is not included.

I would like to add that not all telehealth practice is reported in (gray) articles, but they still might occur with positive or negative implications. To get a view on this the scoping review could have been extended by organizing expert meetings /consultation exercise  according to the approach of  Arksey & O’Malley (2005DOI: 10.1080/1364557032000119616 who writes “a consultation exercise in this sort of study may enhance the results, making them more useful to policymakers, practitioners and service users”

Author Response

Response to Reviewer:

Reviewer 3:

The manuscript is nicely written and highly relevant for the research community in telehealth.

Response: We would like to thank the reviewer for this encouraging comment.

I just have some suggestions to enhance the transparency of the result in the attached document (document attached).

Response: We would like to thank the reviewer for these valuable suggestions. We have addressed all the comments in the attached document and in the updated version of the manuscript.

Furthermore, in the discussion section, you make clear that gray literature is not included.

I would like to add that not all telehealth practice is reported in (gray) articles, but they still might occur with positive or negative implications.

Response: We would like to thank the reviewer for this suggestion. We have now clarified in the discussion section that gray literature has not been included in our review.

To get a view on this the scoping review could have been extended by organizing expert meetings /consultation exercise  according to the approach of  Arksey & O’Malley (2005DOI: 10.1080/1364557032000119616 who writes “a consultation exercise in this sort of study may enhance the results, making them more useful to policymakers, practitioners and service users”

Response: We would like to thank the reviewer for this suggestion. We have now duly referenced Arksey & O’Malley’s important work on scoping reviews. We also now discuss in the limitation section, why we could not pursue this optional step as suggested by Arksey & O’Malley while we have fulfilled the mandatory five stages/steps suggested by them.

Round 2

Reviewer 1 Report

The revision looks much better.

Nice work!